# Local ship speed reduction effect on black carbon emissions measured at remote marine station

Mikko Heikkilä[1], Krista Luoma[1], Timo Mäkelä[1], and Tiia Grönholm[1]

[1]Finnish Meteorological Institute, Helsinki, Finland

**Correspondence:** Mikko Heikkilä (mikko.heikkila@fmi.fi)

**Abstract.** Speed restrictions for ships have been introduced locally to reduce the waves and turbulence causing erosion, and safety hazards, and to mitigate the air and underwater noise emissions. Ship speed restrictions could be used to minimise the climate impact of maritime transport since many air pollutants in ship exhaust gas are reduced when travelling at lower speeds. However, for example, methane and black carbon emissions do not linearly correlate with the load of internal combustion engines. Therefore, the effect of speed restrictions may not be trivial. Black carbon concentrations from ship plumes were examined at the remote marine site in the Finnish Southwestern archipelago. Ships with service speeds over 15 knots and equipped with an exhaust gas cleaning system were analysed for black carbon emissions as a function of speed. Both unadjusted and weather-adjusted main engine loads were modelled to determine load-based emission factors. Black carbon concentration per kilogram of fuel decreased as a function of engine load. However, as calculated per hour the black carbon emission increased as a function of ship speed reaching peak values at around 15-20 knots and decreasing thereafter. In terms of local air quality, total black carbon emission per nautical mile was the highest around the halved speeds, 10-13 knots, or when the speed was higher than 20-23 knots. From a climate warming perspective, the $CO_2$ emissions dominated the exhaust gas and reducing the speed decreased the global warming potential in $CO_2$ equivalent both per hour and per nautical mile.

## 1 Introduction

Speed limits have been discussed widely as a measure to mitigate the effects of underwater noise, air, and water pollution originating from seagoing vessels (e.g. MacGillivray et al., 2019; Woo and Im, 2022). As the resistance of the vessel moving through the water is known to be the largest contributing factor for fuel consumption (Hollenbach, 1998), it is then natural to assume that restricting speeds would directly lead to a reduction in air emissions (Yau et al., 2012). However, the problem is not simple: restricting speed could lead to an increase in the need for carriage and therefore, to an increase in total emissions (Elkafas and Shouman, 2021; Tan et al., 2022). Local speed limits have been introduced in many areas for a variety of reasons. Safety is one of the primary concerns. Also, erosion is increased when ships operate at higher speeds causing wake and turbulence (Almström et al., 2021, 2022; Almström and Larson, 2020; Benassai et al., 2013; Bilkovic et al., 2019; Dam et al., 2008; Roo and Troch, 2015; Gunnel et al., 2014; Houser, 2010; Parnell et al., 2015; Stumbo et al., 1999). Recently, also underwater noise has become a relevant factor when considering vessel speeds (Jalkanen et al., 2022; Lajaunie et al., 2023).

Air emissions from ships can be categorised roughly by their impact on global warming, air quality and the environment. In many cases, focusing only on one could have a negative impact on the other. For example, fuel sulfur content limits have led to increased harmful discharges into the sea (Jalkanen et al., 2024) and uptake of liquefied natural gas as shipping fuel into increased methane emissions (Lindstad and Rialland, 2020). California has successfully implemented voluntary speed limits in the effort of mitigating air pollution from sea-going vessels (Linder, 2018), as many pollutants such as nitrogen oxides ($NO_x$) and particulate matter (PM) are associated with detrimental health and environmental effects (Chen and Hoek, 2020; Nunes et al., 2020; Orellano et al., 2020; Viana et al., 2020; Wang et al., 2020, 2019; Zhang et al., 2021).

Black carbon (BC) emissions from marine engines have been studied widely using various methods to establish accurate emission factors $(\mathrm{g\,BC\,(kg\,fuel)^{-1}})$ for different engine and vessel types (e.g., Cappa et al., 2014; Corbin et al., 2020; Lack et al., 2008; Schlaerth et al., 2021) and the association between engine load and BC emissions has been shown (Jiang et al., 2018; Lack and Corbett, 2012). The Intergovernmental Panel on Climate Change (IPCC) report in 2007 noted that the effect of BC on climate change is probably larger than previously thought (Service, 2008). Black carbon has also been shown to contribute significantly to the health burden caused by fine particulate matter (Chowdhury et al., 2022) and recently an association was made also between lung deposited surface area and BC from ships (Lepistö et al., 2022).

Monitoring and inventorying ship emissions have traditionally been done by bottom-up modelling using ship Automatic Identification System (AIS) data combined with emission factors (e.g. Jalkanen et al., 2009, 2012; Woo and Im, 2021) and very recently also combining meteorological factors in the resistance modelling (Kim et al., 2023). Establishing accurate emission factors and modelling the contribution of weather impact relies on model-testing, remote sensing and measurements conducted in both test-bed conditions and onboard. There still lies a knowledge gap between what is known and what is unknown, especially concerning ships equipped with exhaust gas cleaning systems (EGCS) that are getting more popular since the global reduction of ship fuel sulphur content in 2020. This research aims to answer the question of what is the local effect of restricting speed to ship BC emissions and how aerodynamic resistance caused by wind and the hydrodynamic effect of waves impact modelling the correct main engine load on a ship.

## 2 Data and methods

### 2.1 Measurement site

Exhaust gas plumes originating from passing ships between 26[th] May 2022 - 14[th] October 2023 were examined at the remote sensing site of Utö island (59° 47'N 021° 22'E) situated at the outermost part of the Finnish South-West archipelago (Figure 1). The island is small, its area is 0.81 km$^2$ and its year-round population is less than 40 people. Ships entering and exiting the ports of cities Turku and Naantali pass in proximity (approximately 500 meters) of the Utö island. Moreover, in Utö there is a pilot station to supply pilotage for the archipelago fairway. Pilot boarding speed is normally around 10 knots, which means that vessels with higher operating speeds need to slow down significantly for the pilot boarding or disembarking, and ships that have service speeds close to the boarding speed do not need to alter speed. Some passing vessels are on regular routes to and from the ports supplied by the Utö fairway and their officers carry pilot exemptions. In this case, these ships do not need

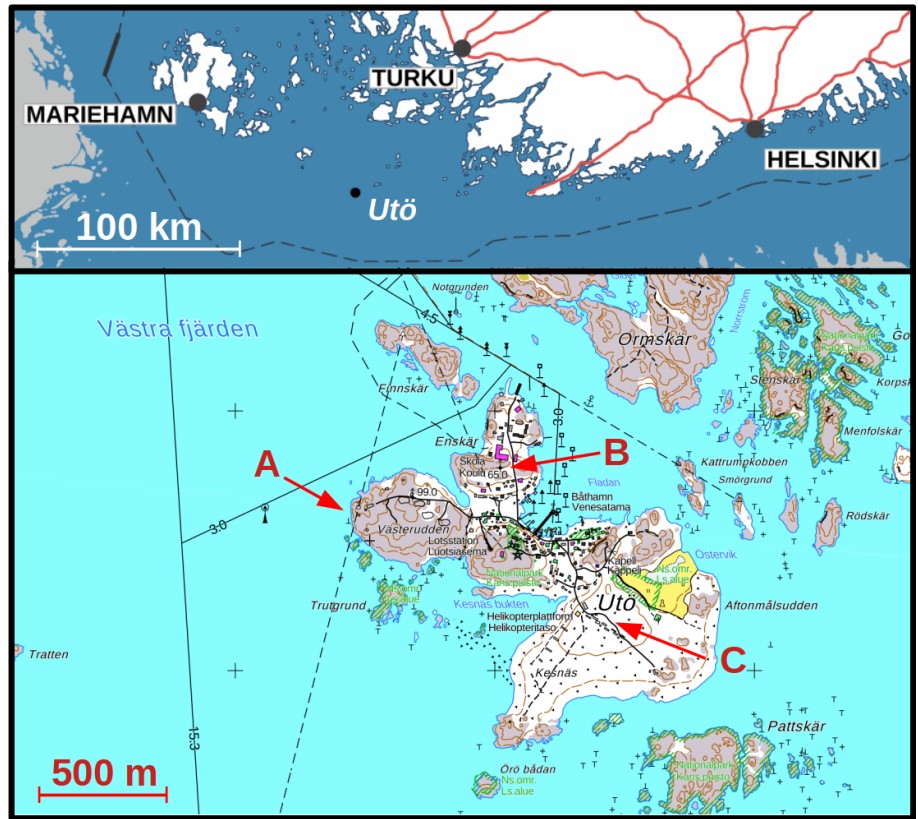

**Figure 1.** Utö location at Finnish south-west archipelago and below a detailed map of Utö island. Measurement locations are shown in the map by three red arrows A) Sea station, B) ICOS-station, and C) Air quality station. Black thin lines present the centre of the shipping fairways. The fairway in the north-west direction on the western side of the island is the main shipping lane in the area. The maximum draught of the fairway for safe navigation is 15.3 m

to slow down at the pilot station. Based on this a dataset was created with as many ships as possible passing the pilot station at various speeds.

A wide range of meteorological, aerosol particle, gas, and biological measurements are carried out in Utö Atmospheric and Marine Research Station owned by Finnish Meteorological Institute (FMI) (Laakso et al., 2018). The station belongs, for example, to the HELCOM marine monitoring network and to the Integrated Carbon Observation System ICOS. The station has three different measurement locations A) Sea station, B) ICOS-station, and C) Air quality station (see the map of Utö, Figure 1).

## 2.2  Ship position data and plume identification

An AIS receiver and an automatic camera are installed in the Sea station. When the AIS signal from the approaching ship is received, the camera starts automatically to take pictures every 30 seconds. Meteorological data (wind velocity and direction,

pressure, and temperature) are measured in the Sea station and synchronized with the AIS data in the file. The plumes were identified manually by checking the AIS information of bypassing ships and possible increases in the BC and $CO_2$ when the wind was from the direction of the nearest shipping lane ($180° - 360°$). Then the start and end times of the plumes were selected based on the BC and $CO_2$ data. The average duration of the plume was about five minutes. Figure 2 shows an example of one ship passing by Utö - data of the ship's location, speed, true heading, and course over the ground were acquired from the AIS data. The plume origin point (a black cross in Figure 2) was determined based on an assumption that the emission plume was transported to the station directly following the wind direction. Therefore, the closest data point from the direction of the wind was selected to represent the speed, true heading and course over the ground of the ship.

## 2.3 Black carbon and carbon dioxide measurements

At the Air Quality Station, BC concentration was measured with an aethalometer (Magee Scientific model AE33), and $CO_2$ concentration with a LI-COR infrared gas analyser (Biosciences model LI-7000). Both of these instruments were installed in the same sample line, which had an inlet on the roof of the measurement cottage roughly at the height of 10 meters from the sea surface (5 meters from the ground level). The sample line had one nafion-dryer installed. The aethalometer and the LI-COR resolutions are 1 min and 5 s, respectively.

BC measurements by an aethalometer are based on filter collection and optical detection. In deriving the BC concentration, a constant mass absorption cross-section, which describes the relation between absorption and BC mass, is assumed. Therefore, based on the recommendations by Petzold et al. (2013), the definition of the measured BC is the so-called equivalent black carbon (eBC). However, for clarity, we use the term BC referring also to the measurements throughout the article. The 880 nm channel with the default mass absorption cross-section of 7.77 $m^2$ $g^{-1}$ of the AE33 were used to acquire the BC concentration. Normally, the AE33 data is corrected with a dual-spot correction algorithm, which corrects for measurement artefacts caused by the filter (Drinovec et al., 2015). However, probably due to too high sample relative humidity in the sample line (despite the nafion dryer), the dual-spot correction did not work optimally. Therefore, instead of the dual-spot correction, a correction algorithm suggested by Virkkula et al. (2007) was applied. The correction factors derived by the algorithm were used as 30-day running medians (e.g. Luoma et al., 2021). With this correction scheme, the aethalometer data was more stable and changes in the filter spot did not cause disturbances in the data.

Both BC and $CO_2$ were converted to dry air and STP conditions (0 °C and 1013.25 hPa) and $CO_2$ mixing ratio was converted to the same mass unit as BC measurement ($\mu g$ $m^{-3}$) using the ideal gas law and the molar mass of $CO_2$:

$$c_{CO_2}\left(\frac{g}{m^3}\right) = \frac{c_{CO_2}(ppm) \cdot P \cdot M_{CO_2}}{RT} \tag{1}$$

where $c_{CO_2}$ (g $m^{-3}$) is the concentration of $CO_2$ in mass units, $c_{CO_2}$ (ppm) is the mixing ratio of $CO_2$, $P$ (Pa) is the pressure, $M_{CO_2}$ (44.01 g $mol^{-1}$) is the molar mass of $CO_2$, $R$ (8.31 kg $m^2$ $s^{-2}$ $K^{-1}$ $mol^{-1}$) is the ideal gas constant and T (K) is the temperature.

Before defining the BC and $CO_2$ concentrations for each plume, the background levels were subtracted from the BC and $CO_2$ data. We applied a method introduced by Ausmeel et al. (2019) to calculate the background, which was defined as the median

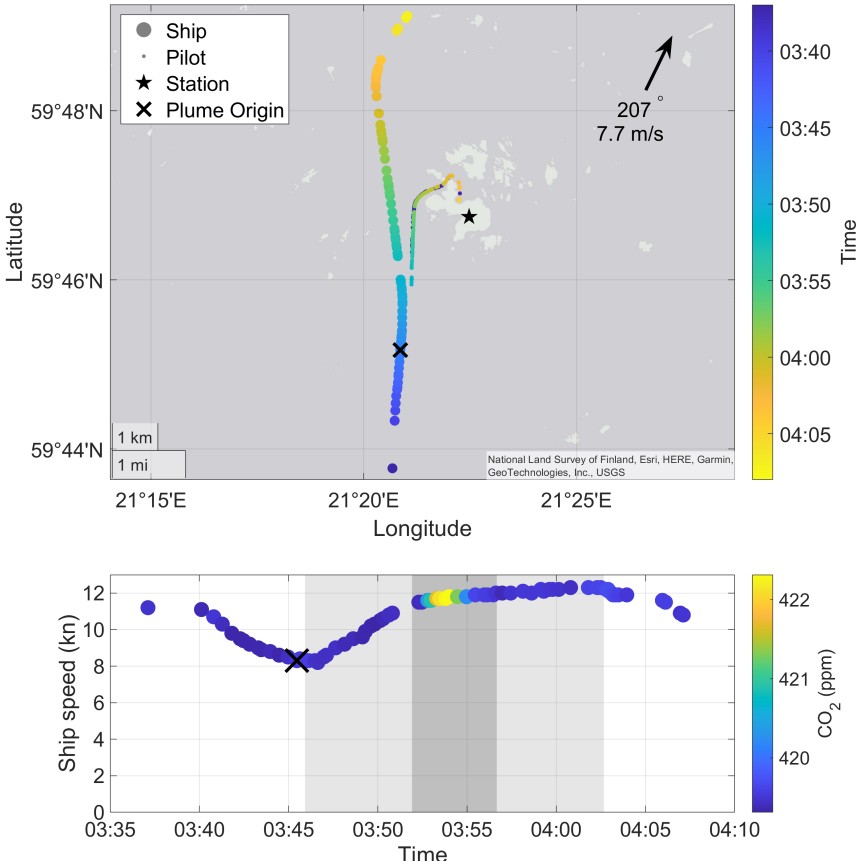

**Figure 2.** An example of retrieving speed, heading and course over ground information of a passing ship. The upper panel shows the wind direction and speed as well as the location of the ship and the pilot vessel. The location of the air quality station is marked with a star. The lower panel shows the ship speed and observed $CO_2$ concentration. The shaded area denotes the time that was used for defining the pollution background levels and the darker shading denotes the time of the actual plume was observed. On both panels, the estimated plume origin is marked with a cross and the colouring of the dots on both panels represents the time.

value of 6 minutes before the plume started and 6 minutes after the plume ended omitting the period of the plume (example in Figure 2). Finally, the dispersed emission $\Delta BC$ and $\Delta CO_2$ were calculated as integral over the duration of the plume. Also, a method, which was developed by Kivekäs et al. (2014) and previously applied at Utö to study the effect of sulphur restrictions on aerosol particle number concentration by Seppälä et al. (2021) was tested. In the method, the background is determined as the $25^{th}$ percentile of 40 consecutive measurements. In our study, where the plumes originated from a relatively short distance (about 2 km) and were short in duration (about 5 min), the method by Ausmeel et al. (2019) produced more

105

well-defined plumes as it took variation in background concentrations better into account. The method by Kivekäs et al. (2014) suits automatic plume detection, whereas the method by Ausmeel et al. (2019) requires more manual work.

Further, the BC emission factor per fuel consumed $EF_{BC}$ can be defined as a dimensionless ratio of $\Delta BC$ and $\Delta CO_2$ times the amount of $CO_2$ in the exhaust, which is derived from the fuel carbon content (FCC, in kg C/kg fuel):

$$EF_{BC}\left(\frac{\text{kg BC}}{\text{kg fuel}}\right) = \frac{\Delta BC}{\Delta CO_2} \times \frac{M_{CO_2}}{M_C} \times FCC\left(\frac{\text{kg C}}{\text{kg fuel}}\right), \tag{2}$$

where $M_{CO_2}$ is the molar mass of $CO_2$ (44.01 g mol$^{-1}$), $M_C$ is the molar mass of carbon (12.01 g mol$^{-1}$). FCC is 0.85 and 0.87 for heavy fuel oil (HFO) and marine gas oil (MGO), respectively (IMO Marine Protection Committee, 2018). Arithmetic mean (AM), geometric mean (GM), median (MED) and standard deviation (SD) were calculated for observed plumes and individual groups of examined vessels.

## 2.4 Engine load, fuel consumption and emission factor calculations

plumes from 47 different ships representing 10 different vessel types were selected for examination. Ships were identified by Automatic Identification System (AIS) data, and ship-specific details were extracted from the IHS Markit ship database. Vessel actual draughts were extracted from AIS data and compared to design draughts from the IHS Markit ship database. Plumes caused by ships with a ratio between actual draught per design draught < 0.9 were considered to be in ballast condition and others in laden condition. Examined vessels are shown in Table 1 and each vessel with their specific details in Table A1 of Appendix A.

The vessels were categorized based on whether they had an exhaust gas cleaning system (EGCS) installed or not. All vessels had medium-speed 4-stroke main engines and were expected to use Heavy Fuel Oil (HFO) if equipped with EGCS and very low sulphur Marine Gas Oil (MGO) if not equipped with EGCS. Ships equipped with EGCS could also operate using MGO and with the EGCS switched off, which could explain possible outliers in the data.

The apparent wind experienced by the vessel was computed using the true wind speed and direction combined with the heading and speed over ground of the vessel recorded from the AIS data using trigonometry (Kim and Yaakob, 2016). The ambient wind data was used to calculate the sea state in the Beaufort scale and the added resistance with resulting speed loss created by sea state was calculated by the method created originally by Townsin and Kwon (1983), revised by the same authors (Townsin and Kwon, 1993; Kwon, 2008).

Aerodynamic and hydrodynamic resistance combined with ship AIS speed data was then used to model the estimated main engine load at the time when the exhaust gas plume data was collected by calculating the resistance through the water using the method created by Hollenbach (1998). Separate resistance constants were used for ships with one or two propellers and ships with and without a bulbous bow. For the calculation of the ship's block coefficient, the method suggested by Jensen (1994) was used to estimate the wet surface of each vessel. Vessel parameters were obtained from the IHS Markit ship database. On ships that have multiple main engines and two propellers, a minimum of two engines were assumed to be online at any time and the maximum engine load before switching on a new engine was set to 90%. For diesel-electric vessels and ships equipped with shaft generators, the estimated auxiliary engine power was added to the main engine power needed for specific speeds. The

**Table 1.** Vessel type, number of vessels (N), number of main engines (N ME) on board, number of propellers (N PR), total main engine power in kW of all main engines fitted on board (ME kW), built year of the ship (BY), ship service speed as per the IHS Markit database (SS), number of examined exhaust gas plumes (PL), number of ships fitted with an exhaust gas cleaning system (EGCS), number of ships with diesel-electric propulsion (DE) and number of ships with shaft generators (SG).

| Vessel type | N | N ME | N PR | ME kW | BY | SS | PL | EGCS | DE | SG |
|---|---|---|---|---|---|---|---|---|---|---|
| Bulk | 1 | 1 | 1 | 6252 | 1995 | 13.5 | 1 | 0 | 0 | 1 |
| Chemical tanker | 5 | 1 | 1 | 4000-9450 | 2004-2011 | 14.0-15.0 | 10 | 0 | 0 | 1 |
| Cruise | 3 | 4-5 | 2 | 32000-55216 | 1993-2020 | 20.0-22.5 | 5 | 2 | 3 | 0 |
| Fish | 1 | 1 | 1 | 827 | 1980 | 12.0 | 2 | 0 | 0 | 1 |
| General cargo | 13 | 1 | 1 | 794-2959 | 1994-2010 | 10.0-14.0 | 37 | 0 | 0 | 12 |
| Other | 2 | 2-3 | 2 | 3600-4440 | 2008-2012 | 13.0 | 2 | 0 | 2 | 0 |
| Product tanker | 6 | 1 | 1 | 4000-8450 | 2004-2021 | 13.0-15.3 | 12 | 1 | 0 | 2 |
| Ropax | 2 | 4-5 | 2 | 29880-32580 | 1991-2001 | 18.5-21.0 | 4 | 1 | 1 | 0 |
| Roro | 13 | 1-2 | 1-2 | 12600-25200 | 2006-2012 | 18.0-22.7 | 138 | 13 | 0 | 13 |
| Tug | 1 | 1 | 1 | 1839 | 1976 | 13.0 | 1 | 0 | 0 | 0 |
| All | 47 | 1-5 | 1-2 | 827-55216 | 1976-2020 | 10.0-22.7 | 211 | 17 | 6 | 30 |

auxiliary engine power ($P_{AE}$) was estimated using the International Maritime Organisation Energy Efficiency Design Index (EEDI) formulas (IMO Marine Protection Committee, 2018).

For ships with total installed main engine power $P_{ME} \geq 10000$ kW:

$$P_{AE} = 0.025 P_{ME} + 250 \text{ kW} \tag{3}$$

And for ships with $P_{ME} \leq 10000$ kW:

$$P_{AE} = 0.05 P_{ME} \tag{4}$$

There were 6 diesel-electric ships with multiple engines among the studied vessels (Cruise 1, 2 and 3, Ropax 1, Other 1 and Other 2), 30 with shaft generators and 11 with neither. Each ship's main engine power was modelled to its service speed + 5 knots. The additional speed was needed as the service speed is typically reached with 80% main engine load, but many ships in the dataset were ice-classed with additional installed main engine power. If the power needed for the speed exceeded the total installed main engine power, the main engine load was set to 100%. A sample of unadjusted modelled engine loads is presented in Figure 3. It is worth noting that not all vessels use 100% main engine load even at service speed + 5 knots in the model. This is probably true in real life, for example, ships with ice class might have excess engine power as per the ice class demands. Also, vessels designed for towing cargo, such as Tug 1 in the dataset, have more main engine power installed than they would need to reach their design service speed + 5 knots (Figure 3).

The AIS data contains a value for the ship's draught, which is fed to the AIS transmitter from the ship. These values were compared to the design draughts of each vessel. If the ratio of actual draught per design draught was less than 0.9, the vessel

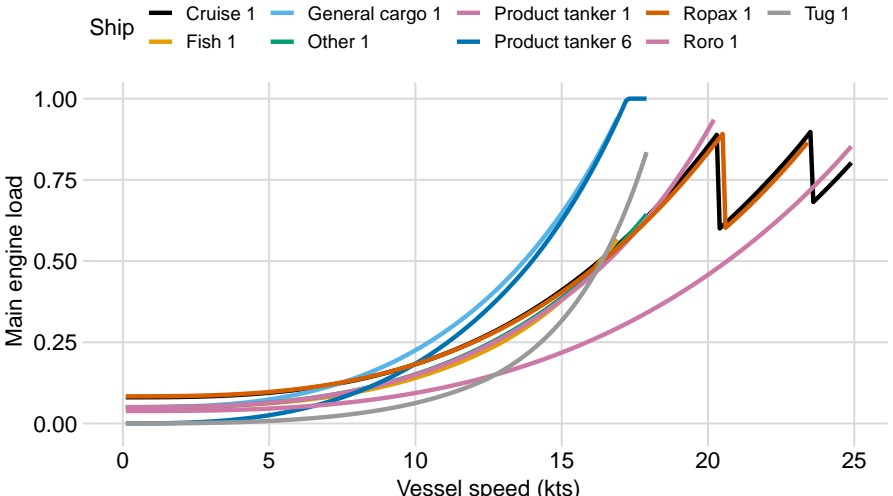

**Figure 3.** Modelled main engine loads without meteorological parameters of a sample of studied vessels with vessel speed over ground in knots on the x-axis and modelled engine load (0-1) on the y-axis. Vessels Cruise 1 and Ropax 1 have multiple main engines, which leads to a decrease in load as a function of speed when an additional engine is started.

was considered to be in ballast condition and otherwise in laden condition, when passing the measuring site (Figure 4). Two vessels, Tug 1 and Fish 1, reported actual draughts that were significantly larger than their design draughts, which could be caused by user error or error in the ship database. Smaller exceeding of the design draught could be because of trim or water density, which is usually around $1.010 \, \text{t m}^{-3}$ in the Baltic Sea and the design draught is calculated for a water density of $1.025 \, \text{t m}^{-3}$. Actual draughts and water density of $1.010 \, \text{t m}^{-3}$ were used in the modelling of resistance through the water.

Modelled main engine load and measured BC values from ship plumes were analysed for correlation and to create statistical models to estimate emissions created by changes in speed and main engine load. The obtained regression formula coefficients for BC output $(\text{g BC} \, (\text{kg fuel})^{-1})$ as a function of engine load were used to model the absolute BC emission $(\text{g BC h}^{-1})$ as a function of speed. For this, the main engine fuel consumption $(F_{\text{ME}})$ of each ship was modelled using a base-specific fuel oil consumption $(\text{SFOC}_{\text{Base}})$ in g kWh$^{-1}$ obtained from the IHS Markit ship database for the specific ship multiplied by the unitless generic relative specific fuel oil consumption $(\text{SFOC}_{\text{Relative}})$ described by Jalkanen et al. (2012):

$$F_{\text{ME}} = \text{SFOC}_{\text{Base}} \times \text{SFOC}_{\text{Relative}}, \tag{5}$$

where

$$\text{SFOC}_{\text{Relative}} = (\alpha L^2 + \beta L + \gamma) \tag{6}$$

where $L$ is the engine load (0-1), $\alpha = 0.45$, $\beta = -0.71$, and $\gamma = 1.28$. BC emission (g h$^{-1}$) was then calculated as:

$$\text{BC}_s \left(\frac{g}{h}\right) = F_{\text{ME}} \left(\frac{\text{g fuel}}{\text{kWh}}\right) \times P_{\text{ME}}(\text{kW}) \times \text{BC}_L \left(\frac{\text{g BC}}{\text{g fuel}}\right) \tag{7}$$

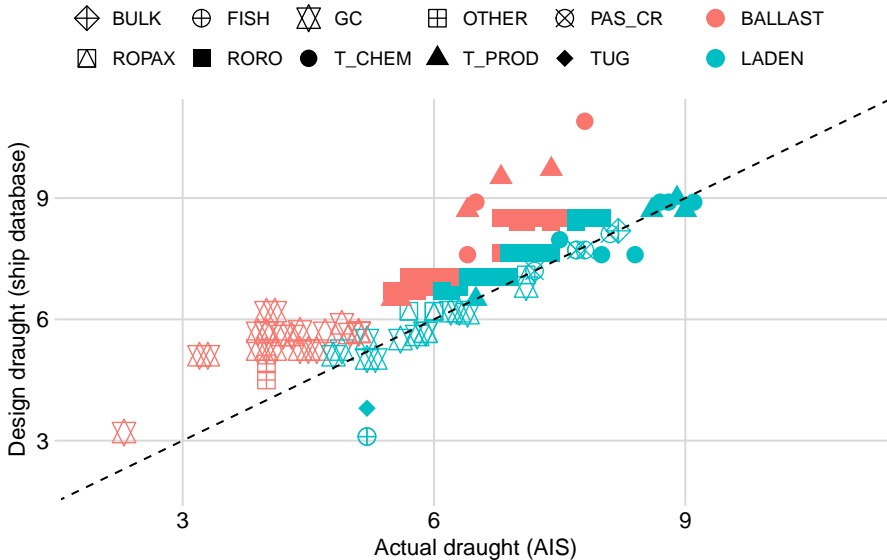

**Figure 4.** Actual draughts of studied vessels (x-axis) compared to design draughts (y-axis) from the IHS Markit ship database in metres. Different vessel types are indicated with corresponding shapes. Vessels in laden condition in green and in ballast condition in orange.

where $BC_s$ is the BC output (g h$^{-1}$) for a specific speed, $F_{ME}$ is the main engine fuel consumption for a specific speed (g fuel kWh$^{-1}$), $P_{ME}$ is the modelled main engine power need for specific speed and $BC_L$ is the load-specific BC output (g BC (g fuel)$^{-1}$).

BC output in g per nautical mile (NM) was calculated as:

$$BC_d\left(\frac{g}{NM}\right) = \frac{BC_s(\frac{g}{h})}{S(\frac{NM}{h})} \tag{8}$$

where $BC_d$ is the BC output in grams per NM, $BC_s$ is the BC output in grams per hour and S is the speed of the vessel in NM per hour.

Vessel total greenhouse gas emissions were calculated using 20 and 100-year global mean warming potential (GWP$_{20}$ and GWP$_{100}$) of BC, estimated as 1600 and 460 (Fuglestvedt et al., 2010; Gasser et al., 2017) in combination with vessel $CO_2$ emissions to determine the effect of speed change in the total GHG as $CO_2$ equivalent ($CO_2$e). Emission factors used for $CO_2$ were 3.11 for HFO and 3.21 for MGO (MEPC, 2021). The code for modelling the engine load was created using Python programming language. The data analysis was performed in R (version 4.2.1) and plotted using the Ggplot2 package (Wickham, 2009).

## 3 Results

### 3.1 Black carbon emission factors

The fuel-based emission factors $(\mathrm{g\,BC\,(kg\,fuel)^{-1}})$ for vessels equipped with an Exhaust Gas Cleaning System (EGCS) were significantly lower (arithmetic mean: 0.22, geometric mean: 0.17, median: 0.15, standard deviation: 0.21) than for vessels without EGCS (AM: 0.99, GM: 0.82, MED: 0.83, SD: 0.68). The statistical significance in the difference of calculated emission factors between EGCS-equipped and vessels without EGCS was confirmed with the Mann-Whitney U-test ($p < 0.01$). There is no statistically significant difference ($p = 0.06$) between BC emissions of ships in ballast condition (AM: 0.50, GM: 0.32, MED:0.29, SD: 0.50) and ships in laden condition (AM: 0.44, GM: 0.24, MED: 0.18, SD: 0.63). The fuel-based emission factors for ships with service speeds > 15.0 knots were significantly lower ($p < 0.01$) (AM: 0.25, GM: 0.18, MED: 0.15, SD: 0.24) than ships with service speeds $\leq$ 15.0 knots (AM: 1.06, GM: 0.88, MED: 0.86, SD: 0.71). $\mathrm{EF_{BC}}$ as a function of ship service speed between different loading conditions and EGCS is presented in Figure B1 of Appendix B. Vessel type BC emission factors are presented in Table 2 and Figure 5.

**Table 2.** Vessel type, BC emission factor ($\mathrm{EF_{BC}}$) in the unit $\mathrm{g\,BC\,(kg\,fuel)^{-1}}$ arithmetic mean (AM), geometric mean (GM), median (MED), standard deviation (SD), and number of examined vessels ($N$). There was only one bulk carrier and one tug with one plume each in the dataset and therefore standard deviation could not be calculated.

| Vessel type | AM | GM | MED | SD | $N$ |
|---|---|---|---|---|---|
| Bulk | 0.20 | 0.20 | 0.20 | na | 1 |
| Chemical tanker | 0.72 | 0.59 | 0.79 | 0.40 | 10 |
| Cruise | 0.31 | 0.26 | 0.29 | 0.21 | 5 |
| Fish | 1.18 | 1.13 | 1.18 | 0.50 | 2 |
| General cargo | 1.18 | 1.05 | 1.15 | 0.63 | 37 |
| Other | 0.38 | 0.37 | 0.38 | 0.04 | 2 |
| Product tanker | 0.75 | 0.73 | 0.74 | 0.18 | 12 |
| Ropax | 0.37 | 0.23 | 0.15 | 0.47 | 4 |
| Roro | 0.22 | 0.17 | 0.15 | 0.20 | 137 |
| Tug | 3.91 | 3.91 | 3.91 | na | 1 |
| All vessels | 0.48 | 0.28 | 0.24 | 0.56 | 211 |

### 3.2 Correlation analysis

Vessel speed over ground correlates negatively with the BC emission factor. Pearson's correlation between speed and BC emission factor on ships with EGCS was $-0.60$ (95% confidence interval $-0.70$–$-0.49$, $p > 0.01$) and on ships without EGCS it was $-0.32$ (95% confidence interval $-0.51$–$-0.09$, $p = 0.01$). The correlation between speed over ground and BC emission factor on ships with service speeds > 15.0 knots was $-0.69$ (95% confidence interval $-0.76$–$-0.59$, $p < 0.01$) and on ships

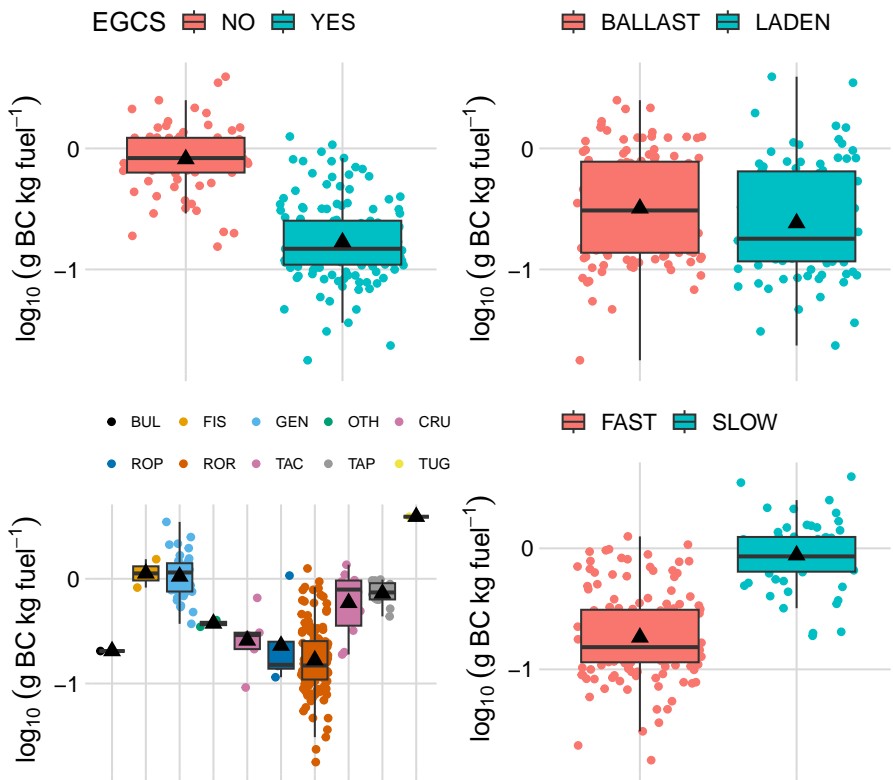

**Figure 5.** Boxplots of the emission factor $\log_{10} \left( \mathrm{g\,BC\,(kg\,fuel)^{-1}} \right)$ measured from passing ships grouped by having an Exhaust Gas Cleaning Systems (top left), by being in ballast or laden condition (top right), having a service speed larger or less than 15.0 knots (bottom right) or by vessel type (bottom left) showing median as black line with interquartile ranges, outliers and arithmetic means with black triangles. BUL: bulk carrier, FIS: fishing vessel, GEN: general cargo vessel, OTH: other vessel type, CRU: cruise passenger vessel, ROP: ropax vessel, ROR: roro vessel, TAC: chemical tanker, TAP: product tanker, TUG: tug.

with service speeds $\leq 15.0$ knots $-0.21$ (95% confidence interval $-0.44$–$0.05$, $p = 0.12$). The correlation between speed over ground and BC emission factor on ships in ballast condition was $-0.72$ (95% confidence interval $-0.80$––$0.62$, $p < 0.01$) and on ships in laden condition it was $-0.63$ (95% confidence interval $-0.74$––$0.49$, $p < 0.01$).

Tug1 which had only one exhaust gas plume in the dataset and passed the measuring site at a very low speed has an emission factor 4 times larger than the mean for vessels without EGCS. A visual confirmation from the automatic camera at the measuring site confirmed that Tug1 was towing a barge while passing Utö. As modelling the engine load was impossible, Tug1 was removed from further analysis, leaving 210 plumes in the dataset. All except for one vessel passing the measuring site with a speed $> 15$ knots over the ground were equipped with EGCS. Most vessels without EGCS were also ships with lower service speeds and therefore would not need to slow down for the pilot exchange. They also emit more BC than the

EGCS-equipped vessels and therefore create bias in the analysis if included. Further analyses were performed only for the EGCS-equipped vessels using 142 exhaust gas plumes and three different vessel types in the dataset.

### 3.3 Emission factor as a function of engine load

Visual examination combined with knowledge from previous literature confirmed a non-linear relationship between modelled engine load and BC emission factor. To define the emission factor of BC as a function of engine load, $2^{nd}$ degree polynomial regression was fitted to the logarithm of observed and modelled values. Adjusting for meteorological parameters had a small effect when fitting the regression. In the exhaust gas plumes emitted by EGCS-equipped vessel, the correlation between unadjusted modelled engine load and BC emission factor was $-0.61$ (95% confidence interval $-0.70 - -0.50$, $p < 0.01$)

and the goodness of fit (adjusted $r^2$) of the polynomial regression was 0.46. When the engine load was adjusted for weather conditions, the correlation between engine load and BC emission factor was -0.61 (95% confidence interval -0.69 - -0.48, $p > 0.01$) and the adjusted $r^2$ of the polynomial regression was 0.41. The EGCS dataset was dominated by plumes measured from roro-type vessels (137 observations) and the adjusted $r^2$ for roro-type vessels only was 0.39. Two plumes were observed from EGCS-equipped cruise vessels, each from a different vessel. For vessel Cruise 3 the model fits well (observed $EF_{BC}$:

0.21, predicted 0.25) but poorly for vessel Cruise 2 (observed $EF_{BC}$: 0.09, predicted: 0.29). Three plumes were observed from EGCS-equipped vessel Ropax 1 on which the model predicts well on higher engine load (observed $EF_{BC}$: 0.11 and 0.15, predicted: 0.13 and 0.14) but poorly with lower engine load: (observed: 1.07, predicted: 0.31). The obtained load-based emission factor formula for the BC output from EGCS-equipped vessels is:

$$\log_{10}(\text{LEF}_{\text{BC}}) = 2.10L^2 - 2.85L - 0.02 \tag{9}$$

where $\text{LEF}_{\text{BC}}$ is the load-based BC emission factor (g kg fuel$^{-1}$) and $L$ is the engine load (0-1) (Figure 6).

### 3.4 Total greenhouse gas emissions at various speeds

As shown above, the fuel consumption-based BC emissions from ships that are equipped with EGCS are dependent on the engine load. As the engine load varies with different power demands, which again is dependent on the speed of the vessel, BC emissions can be modelled as a function of vessel speed. Five different ships representing various vessel types were chosen

as a sample from the studied vessels. Three of them have conventional propulsion systems (Roro1, Roro2 and Tanker), which means the main engines are connected mechanically to the propeller shaft. The other two (Ropax and Cruise) are diesel-electric, which means the main engines are connected to generators, and propellers are rotated with electric motors. 80% engine load was chosen as cut-off, as the data from our observations was limited to around 80% modelled engine load. Chosen vessels also had different specific fuel oil consumption (SFOC$_{\text{Base}}$) based on the ship database information varying from 180 to 220

240  g kWh$^{-1}$.

BC emission was modelled as grams per hour and as grams per nautical mile to distinguish between service speed and reduced speed. Due to the parabolic models for fuel consumption and BC emission factor curves, BC emissions increase non-linearly from when the ship starts moving until reaching the first peak, which seems to be at around 10-12 knots speed. BC

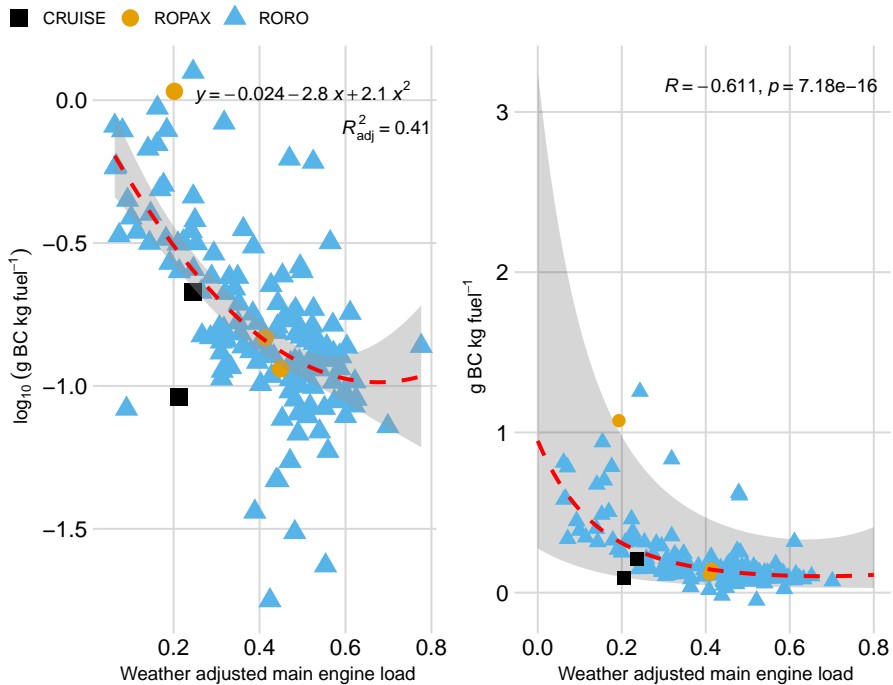

**Figure 6.** Left: scatterplot of $\log_{10}$ BC emission factor (g kg fuel$^{-1}$) as a function of weather adjusted main engine load with $2^{nd}$ degree polynomial regression (red dashed line) and 95% confidence interval (grey area). Right: BC emission factor (g kg fuel$^{-1}$) with the same regression (red dashed line) and 95% prediction interval (grey area).

emissions then decrease until reaching similar levels than at 5-knot speed. This seems to happen at around the ship service speed, from where the BC emissions seem to increase again. Vessel service speed is reached where main engine fuel consumption is around $\text{SFOC}_{\text{Base}}$ this being typically around 60-80% load for most marine engines. Within the modelled vessels, this would be at around 15-20 knots speed. Increasing speed further also increases the BC emissions non-linearly. $CO_2$e emissions from black carbon represent on average 15.5% of total greenhouse gases using $\text{GWP}_{20}$ and 5.2% using $\text{GWP}_{100}$. As $CO_2$ dominates the total greenhouse gas emissions, they are reduced non-linearly with a reduction of speed (Figure 7).

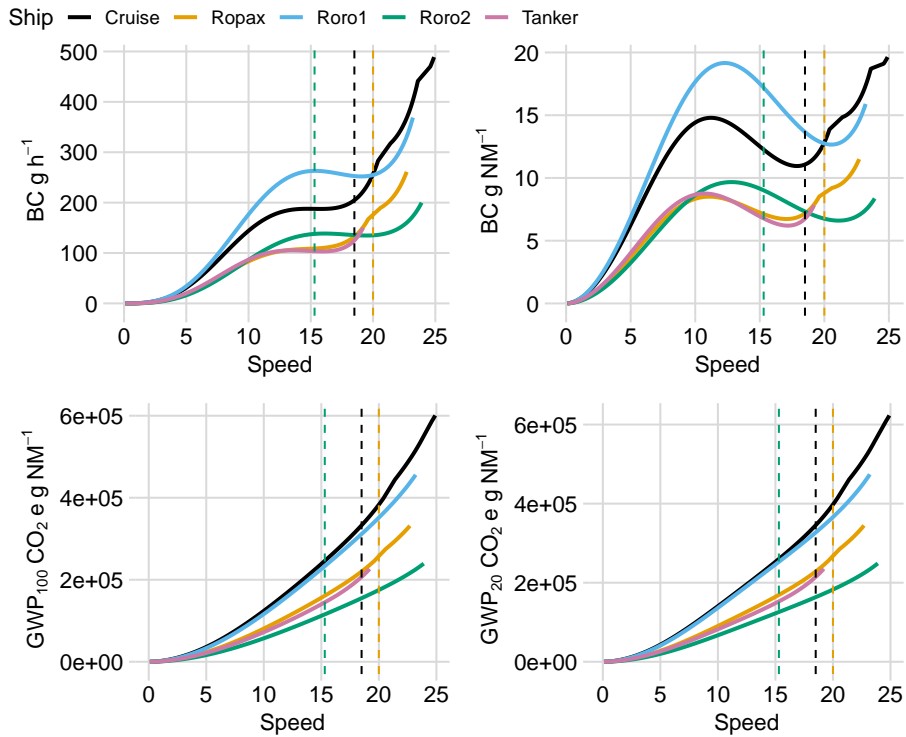

**Figure 7.** BC emission (g h$^{-1}$) as a function of speed of 5 different modelled ships (top left), BC emissions (g per nautical mile) of the same vessels as a function of speed (top right), total greenhouse gas emissions ($CO_2$ + BC, in g per nautical mile) using global warming potential for 100 years for BC (bottom left) and with global warming potential for 20 years (bottom right). Vessel service speeds as vertical dashed lines. Vessels Cruise, Roro1 and Roro2 have the same service speed (20.0 knots).

## 4 Discussion

The average BC emission factors (arithmetic mean: 0.48, geometric mean: 0.28, median: 0.24 and standard deviation: 0.56 g BC (kg fuel)$^{-1}$) derived from the aethalometer measurements for all examined ship exhaust gas plumes and vessel type-specific means (Table 2) are in line with the results from previous literature using various measuring methods: Schlaerth et al. (2021) measured 78 plumes using a custom-built light absorption photometer obtaining a geometric mean emission factor of $0.49 \pm 0.62$ g BC (kg fuel)$^{-1}$ from onshore measurements of various harbour crafts, Cappa et al. (2014) used a single particle photometer and a soot-particle aerosol mass spectrometer to calculate a weighted average of $0.23 \pm 0.15$ g BC (kg fuel)$^{-1}$ for the research vessel *Miller Freeman*. Buffaloe et al. (2014) measured 91 ship plumes from onboard a research vessel combining multiple techniques (photoacoustic spectrometer and particle soot absorption photometer for light absorption, laser-induced incandescence and mass spectrometry) obtaining a geometric mean for all ships of $0.31 \pm 0.31$ g BC (kg fuel)$^{-1}$, $0.26 \pm 0.26$ for ships with slow-speed diesel engines (SSD), $0.27 \pm 0.12$ for ships with medium-speed diesel engines (MSD) and $0.32 \pm 0.26$ for ships with high-speed diesel engines (HSD). Lack et al. (2008) measured 101 ship plumes from onboard a research vessel using

a photoacoustic technique to calculate the light absorbing carbon obtaining mean emission factors $0.41 \pm 0.27 \, \mathrm{g\,BC\,(kg\,fuel)^{-1}}$ for SSD-powered ships, $0.97 \pm 0.66$ for MSD-powered ships and $0.36 \pm 0.23$ for HSD-powered ships with vessel-specific emission factors: $0.38 \pm 0.27$ for tankers, $0.80 \pm 0.23$ for containerships, $0.40 \pm 0.23$ for cargo carriers, $0.38 \pm 0.16$ for bulk carriers, $0.97 \pm 0.66$ for tug boats and $0.36 \pm 0.23$ for passenger boats. Petzold et al. (2008) used a particle soot absorption photometer and measured $0.17 \pm 0.04 \, \mathrm{g\,BC\,(kg\,fuel)^{-1}}$ from a single containership with aeroplane measurements, Sinha et al. (2003) used an aeroplane to collect filter samples from plumes of a tanker and a containership. The filters were analysed with an optical transmission technique obtaining a mean emission factor of $0.18 \pm 0.02 \, \mathrm{g\,BC\,(kg\,fuel)^{-1}}$.

As seen above, BC concentration can be measured by various methods that rely on the different properties of the BC (e.g, chemical composition, refractivity or optical properties), they are based on different assumptions (e.g., mass absorption cross-section, calibrations), and they are sensitive to different measurement artefacts (errors caused by e.g., relative humidity, filter fibres, other absorbing substances) (Petzold et al., 2013). A study by Aakko-Saksa et al. (2022) investigated various methods to measure BC emissions in a laboratory and on-board. In their results, BC concentration measured by an aethalometer showed about 1.26 times more than the other used methods (filter smoke number, laser-induced incandescence, photoacoustic spectroscopy). On an on-board BC measurement comparison by Cappa et al. (2014) the laser-induced incandescence showed somewhat lower results to other methods (optical, photoacoustic). Also, Buffaloe et al. (2014) observed that laser-induced incandescence measurements resulted in lower BC emission factors than those measured by optical means on a plume-chasing measurement campaign. Not only did the method to measure BC vary, but also the measurement set-up varied; some studies measured the in-stack emissions of a certain ship and others measured the emissions of various ships by either chasing the plumes or measuring the plumes from passing ships. Even though BC is an inert compound and its chemical composition is not expected to change, it can still get coated with other materials inside the scrubber or during ageing that affect the optical properties of BC particles. For example, purely scattering or slightly absorbing coating can increase the light absorption of the coated BC particle (Bond et al., 2006; Lack and Cappa, 2010). This so-called lensing effect can lead to an overestimation of BC concentration when BC is derived by optical methods and a constant mass absorption cross section (for primary aerosol) is used.

Here, we assumed that the BC particle ageing has only a minor effect on the optically derived BC concentration. The lag time between the plume emission from the stack and detection by the analyzer was less than 5 minutes on average. Only a few of the plumes travelled more than 10 minutes before detection. There was no correlation between the $\mathrm{EF_{BC}}$ and the plume age during this short time period in the marine background station. Even on longer time scales, previous studies have observed the low potential for new particulate material formation in photo-oxidation processes in the exhaust of EGCS-equipped ships (Karjalainen et al., 2022) and for ship emissions in SECA area (Ausmeel et al., 2020). Previously, at Utö, Seppälä et al. (2021) suggested diminished photochemical ageing of the plumes with stricter fuel sulphur restrictions. It is also justified to assume that coating of particles inside the EGCS is not dependent on ship speed or engine load, and therefore conclusions are not biased. Still, in the ECGSs the exhaust cools down and is exposed to humid conditions, which could increase the coating on BC particles in comparison to exhaust that is not treated with an ECGS. Increase in coating could lead to increase in optically measured BC concentration for ECGS plumes compared to plumes from ships without an ECGS. The enhancement in measured

absorption (i.e., BC concentration) due to coating depends on the wavelength, the coating material, the size of the BC core, and the thickness of coating (Lack and Cappa, 2010). For example, for ambient aerosol at an urban measurement site in Barcelona, depending on the amount of coating material, the absorption was increased by 1.1 - 1.6 times at 880 nm (Yus-Díez et al., 2022). With the current data set in this study, it is not possible to estimate whether the absorption was enhanced significantly with the plume aging or between plumes from ships equipped with or without an ECGS.

In the studies mentioned above that observed BC emissions from ships, the ship fleet varied a lot and most of the studies included ships with different service speeds and that operated with low sulphuric fuels. Here, the focus was on ships with service speeds over 15 knots and that were equipped with an EGCS. As the use of EGCSs has been increasing, it underlines the importance of studying also their emissions.

Ships without EGCS had a median BC emission factor 5-fold larger than vessels with EGCS. Previous literature confirms that EGCS reduces particle mass and BC output (Lack and Corbett, 2012; Fridell and Salo, 2014; Lehtoranta et al., 2019). However, in Winnes et al. (2020) and Järvinen et al. (2023) BC output was higher when combusting HFO in combination with EGCS compared to combusting low-sulfur fuel oil without EGCS. The modelled engine load for the ships without EGCS was relatively low in this study, which could explain why their averaged BC output was high. Also, the specific fuel type and chemical composition were not available as the measurements were conducted remotely. The dataset used was dominated by EGCS-equipped vessels, for which reason all further analyses were focused only on EGCS vessels to avoid bias in the results. Therefore, no conclusions can be made concerning ships without EGCS from this study. Further research is needed to determine the load-based emission factor formula for slow-moving vessels without EGCS. The load-based BC prediction model performed reasonably and some outliers could be explained by the ship momentarily slowing down for pilot boarding or disembarking and having more engines online than would be optimum for the speed.

Meteorological parameters have a significant effect on the resistance experienced by the vessel while navigating and they should be taken into account when modelling engine load based on AIS data. As there is no tidal flow at the research site, the effect of currents was omitted in the modelling. The speed penalty calculation developed by Kwon (2008) which was used in this study classifies ships by their Froude number and whether they are in ballast or laden condition. A distinction is also made between container ships and other vessel types. A similar method developed by Jalkanen et al. (2009) was also tried, but it rendered results that were less accurate than Kwon's method. Using actual reported draughts instead of the design draught from a ship database adds to the accuracy of the modelling as it can be used to differentiate ships in ballast and laden conditions.

Most of the vessels in this study were either diesel-electric or were equipped with shaft generators. Therefore, the estimated auxiliary power was added to the modelled main engine power and contributed to the modelled main engine load. However, as it was impossible to distinguish if the shaft generators were in use or not, the measured plumes were a mix of the main engine and possible auxiliary engine exhaust gases. We estimate, that this should not cause significant bias in the results as the vessels with the largest auxiliary demand (cruise and ropax vessels) were mostly also diesel-electric and not equipped with separate auxiliary engines.

 **5 Conclusions**

The median black carbon emission factor ($EF_{BC}$) for 47 ships representing 10 different vessel types measured from a remote marine station was $0.24\,\mathrm{g\,BC\,(kg\,fuel)}^{-1}$. For ships equipped with EGCS it was 0.15 and for ships without EGCS 0.83. $EF_{BC}$ has a strong negative correlation with speed and engine load, which can be modelled to a reasonable degree of accuracy. The majority of vessels equipped with EGCS also had faster service speeds. Based on the results of this study, reducing vessel speed will result in a reduction of greenhouse gas emissions at least for EGCS-equipped vessels powered by fuel oil. However, local speed restrictions might not be beneficial: if speed is increased during the remaining voyage the overall GHG emissions might be more than without the speed restriction. Also, as BC emissions have other effects, such as on human health, local BC concentrations are increased with small reductions in speed. Based on the results, speed limits need to be 10 knots or less for the BC emissions to be the same as with the ship's normal service speeds. This should be considered carefully at locations where the human population are exposed to ship exhaust gas plumes.

**Appendix A: Appendix A**

Table A1: Vessel type, number of main engines (ME) on board, number of propellers (PR), total main engine power in kW of all main engines fitted on board (kW), built year of the ship (BY), ship service speed as per the IHS Markit database (SS), number of examined exhaust gas plumes (PL), if the ship was fitted with an exhaust gas cleaning system (EGCS) or not, if the ship has diesel-electric propulsion or not (DE) and if the ship has a shaft generator or not (SG). The vessel marked with * did not have a service speed entry in the ship database and the value was estimated by comparing it to a similar vessel.

| N | Vessel | ME | PR | kW | BY | SS | PL | EGCS | DE | SG |
|---|---|---|---|---|---|---|---|---|---|---|
| 1 | Bulk 1 | 1 | 1 | 6252 | 1995 | 13.5 | 1 | No | No | Yes |
| 2 | Chemical tanker 1 | 1 | 1 | 4320 | 2009 | 14.6 | 1 | No | No | Yes |
| 3 | Chemical tanker 2 | 1 | 1 | 5800 | 2008 | 14.0 | 4 | No | No | No |
| 4 | Chemical tanker 3 | 1 | 1 | 9450 | 2004 | 14.5 | 1 | No | No | No |
| 5 | Chemical tanker 4 | 1 | 1 | 4320 | 2006 | 15.0 | 3 | No | No | No |
| 6 | Chemical tanker 5 | 1 | 1 | 4000 | 2011 | 14.1 | 1 | No | No | No |
| 7 | Cruise 1 | 5 | 2 | 34560 | 1993 | 20.0 | 3 | No | Yes | No |
| 8 | Cruise 2 | 5 | 2 | 55216 | 2000 | 22.5 | 1 | Yes | Yes | No |
| 9 | Cruise 3 | 4 | 2 | 32000 | 2020 | 20.0 | 1 | Yes | Yes | No |
| 10 | Fish 1 | 1 | 1 | 827 | 1980 | 12.0 | 2 | No | No | Yes |
| 11 | General cargo 1 | 1 | 1 | 2400 | 2001 | 12.0 | 4 | No | No | Yes |
| 12 | General cargo 2 | 1 | 1 | 1650 | 1994 | 11.0 | 7 | No | No | Yes |
| 13 | General cargo 3 | 1 | 1 | 1360 | 1998 | 11.0 | 2 | No | No | Yes |

| N | Vessel | ME | PR | kW | BY | SS | PL | EGCS | DE | SG |
|---|--------|----|----|----|----|----|----|------|----|----|
| 14 | General cargo 4 | 1 | 1 | 2400 | 1997 | 12.0 | 10 | No | No | Yes |
| 15 | General cargo 5 | 1 | 1 | 794 | 2000 | 10.0 | 1 | No | No | Yes |
| 16 | General cargo 6 | 1 | 1 | 1492 | 2000 | 10.7 | 3 | No | No | Yes |
| 17 | General cargo 7 | 1 | 1 | 1800 | 2008 | 12.0 | 3 | No | No | Yes |
| 18 | General cargo 8 | 1 | 1 | 2000 | 1994 | 12.5 | 1 | No | No | Yes |
| 19 | General cargo 9 | 1 | 1 | 1800 | 1999 | 12.5 | 1 | No | No | Yes |
| 20 | General cargo 10 | 1 | 1 | 2700 | 2005 | 13.0 | 1 | No | No | Yes |
| 21 | General cargo 11 | 1 | 1 | 2460 | 1997 | 13.5 | 1 | No | No | Yes |
| 22 | General cargo 12 | 1 | 1 | 2880 | 2002 | 14.0 | 2 | No | No | Yes |
| 23 | General cargo 13 | 1 | 1 | 2959 | 2010 | 14.0* | 1 | No | No | No |
| 24 | Other 1 | 2 | 2 | 4440 | 2008 | 13.0 | 1 | No | Yes | No |
| 25 | Other 2 | 3 | 2 | 3600 | 2012 | 13.0 | 1 | No | Yes | No |
| 26 | Product tanker 1 | 1 | 1 | 8450 | 2005 | 15.3 | 6 | Yes | No | Yes |
| 27 | Product tanker 2 | 1 | 1 | 4000 | 2012 | 13.0 | 2 | No | No | No |
| 28 | Product tanker 3 | 1 | 1 | 4000 | 2012 | 13.0 | 1 | No | No | No |
| 29 | Product tanker 4 | 1 | 1 | 5700 | 2021 | 14.0 | 1 | No | No | No |
| 30 | Product tanker 5 | 1 | 1 | 8450 | 2004 | 15.3 | 1 | No | No | Yes |
| 31 | Product tanker 6 | 1 | 1 | 6000 | 2018 | 13.0 | 1 | No | No | No |
| 32 | Ropax 1 | 5 | 2 | 29880 | 2001 | 18.5 | 3 | Yes | Yes | No |
| 33 | Ropax 2 | 4 | 2 | 32580 | 1991 | 21.0 | 1 | No | No | No |
| 34 | Roro 1 | 2 | 2 | 18900 | 2002 | 20.0 | 9 | Yes | No | Yes |
| 35 | Roro 2 | 2 | 2 | 25200 | 2007 | 22.7 | 5 | Yes | No | Yes |
| 36 | Roro 3 | 2 | 2 | 20000 | 2012 | 20.0 | 25 | Yes | No | Yes |
| 37 | Roro 4 | 2 | 2 | 20000 | 2012 | 20.0 | 24 | Yes | No | Yes |
| 38 | Roro 5 | 2 | 2 | 20000 | 2012 | 20.0 | 6 | Yes | No | Yes |
| 39 | Roro 6 | 2 | 2 | 20000 | 2012 | 20.0 | 4 | Yes | No | Yes |
| 40 | Roro 7 | 1 | 1 | 12600 | 2000 | 20.0 | 6 | Yes | No | Yes |
| 41 | Roro 8 | 2 | 2 | 25200 | 2006 | 22.7 | 8 | Yes | No | Yes |
| 42 | Roro 9 | 2 | 2 | 12000 | 2017 | 18.0 | 21 | Yes | No | Yes |
| 43 | Roro 10 | 2 | 2 | 25200 | 2008 | 22.0 | 7 | Yes | No | Yes |
| 44 | Roro 11 | 2 | 2 | 25200 | 2009 | 22.0 | 4 | Yes | No | Yes |
| 45 | Roro 12 | 2 | 2 | 25200 | 2007 | 22.7 | 13 | Yes | No | Yes |
| 46 | Roro 13 | 2 | 2 | 25200 | 2006 | 22.7 | 5 | Yes | No | Yes |
| 47 | Tug 1 | 1 | 1 | 1839 | 1976 | 13.0 | 1 | No | No | No |

## Appendix B: Appendix B

*Author contributions.* Conceptualization: M.H., T.M. Data curating: T.M., K.L. Formal Analysis: M.H., T.M., K.L. Investigation: T.M, K.L. Methodology: M.H. Project administration: M.H., T.G. Resources: T.M. Software: M.H., K.L. Supervision: T.G. Validation: M.H., T.G. Visualization: M.H. Writing original draft: M.H., T.G. Writing – review and editing: All authors.

*Competing interests.* The authors declare that they have no conflict of interest

*Disclaimer.* This work reflects only the authors' view and CINEA is not responsible for any use that may be made of the information it contains.

*Acknowledgements.* This research was produced as part of the European Union project "EMERGE: Evaluation, control, and mitigation of the environmental impacts of shipping emissions". The EMERGE project has received funding from the European Union's Horizon 2020 – Research and Innovation Framework Programme action under grant agreement No 874990. Observations and research support at Utö Atmosphere and Marine Research Station are partly funded by Integrated Carbon Observing System (ICOS) and the Finnish Marine Research Infrastructure (FINMARI). Special thanks are due to Ismo and Brita Willström for their valuable work in maintenance of the measurements at Utö.

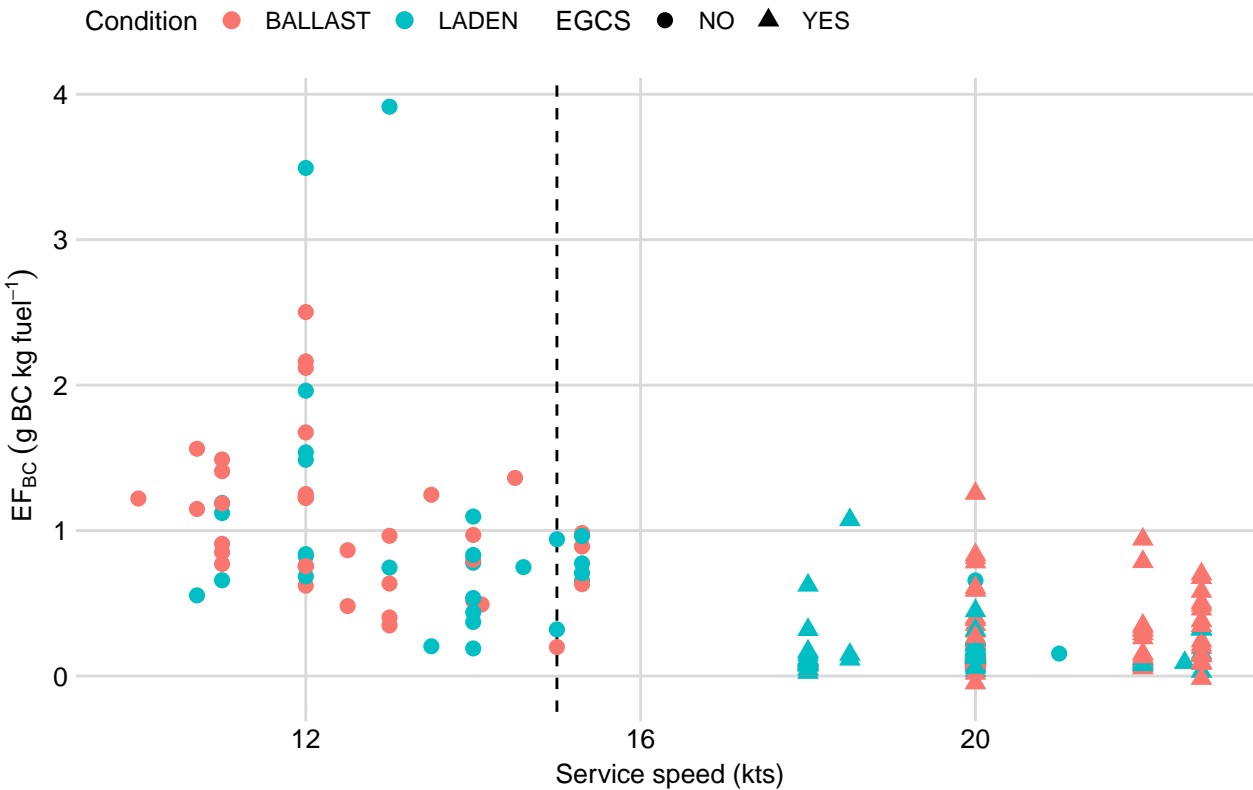

**Figure B1.** Black carbon emission factor $(\mathrm{g\,BC\,(kg\,fuel)^{-1}})$ as a function of ship service speed in knots with loading condition with corresponding colours and if the ship is equipped or not with Exhaust Gas Cleaning System with corresponding symbols. Service speed of 15.0 knots marked with a vertical dashed line.

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
