# Peer review of "Local ship speed reduction effect on black carbon emissions measured at remote marine station"

_EGUsphere, 2023_

## Author Response (AR1)

Replies to reviewers

**Reviewer 1**

In their study, Heikkilä et al. present new measurements of CO2 and black carbon emissions from the plumes of 47 different ships observed at remote station. The measured emission factors are examined in relation to ship speed as well as modelled engine load. The BC and CO2 equivalent emissions are then modelled as function of ship speed and it is shown that ship speed reduction can increase BC emissions but results in reduction of total CO2 equivalent emissions. In general, the study is well executed and adds knowledge to the discussion related to the climate impacts of speed reductions / slow steaming of ships. The main concern is related to the conclusions made on the effect of EGCS systems on the BC emissions from ships. Attached below are my main remarks and technical comments related to the manuscript.

Main remarks:

p8. L139 Under methods, the authors explain that each ship's main engine power was modelled to its service speed + 5 knots. For the reader, it could be clarified why the additional constant 5 knots were added to the service speed

**Reply**: The following was added to the manuscript: "The additional speed was needed as the service speed is typically reached with 80% main engine load, but many ships in the dataset were ice-classed with additional installed main engine power."

p9 Fig. 3 Looking at Figure 3, for two of the modelled ships, Cruise 1 and Other 1, the main engine load decreases when certain vessel speed is reached. The reasoning for this could be discussed in the relevant section, can it be concluded that the vessels start utilizing more than one main engine at certain speed?

**Reply**: This is explained in the manuscript on lines 128-130:

"On ships that have multiple main engines and two propellers, a minimum of two engines were assumed to be online at any time and the maximum engine load before switching on a new engine was set to 90%."

To address this also in the figure, the following was added to the caption of Figure 3:

"Vessels Cruise 1 and Ropax 1 have multiple main engines, which leads to a decrease in load as a function of speed when an additional engine is started."

p11. L178-179 The authors state that "Vessels equipped with Exhaust Gas Cleaning System (EGCS) emit significantly less BC (mean: 0.22 g BC (kg fuel)−1, standard deviation: 0.21) than vessels without EGCS (mean: 0.99 g BC (kg fuel)−1, SD: 0.68)." This reduction in BC in vessels having EGCS compared to ones without EGCS is significant. In previous studies

considering stack measurements a high 89% reduction has also been found (Fridell & Salo, 2014) but other studies considering the effect on scrubber on BC levels measured from stack have reported somewhat lower BC removal efficiencies of 0-37% (Winnes et al., 2020) and 30-40% (Järvinen et al., 2023).

**Reply**: We appreciate pointing this out. Winnes et al., 2020 and Järvinen et al., 2023 both show that BC output is higher when combusting HFO in combination with EGCS compared to combusting low-sulfur fuel oil. We have edited the manuscript as follows:

Discussion (lines 284-289):

"Ships without EGCS had a median BC emission factor 5-fold larger than vessels with EGCS. Previous literature confirms that EGCS reduces particle mass and BC output (Lack et al., 2012, Fridell & Salo, 2014 , Lehtoranta et al., 2019). However, in Winnes et al, 2020 and Järvinen et al, 2023 BC output was higher when combusting HFO in combination with EGCS compared to combusting low-sulfur fuel oil without EGCS. The modelled engine load for the ships without EGCS was relatively low in this study, which could explain why their averaged BC output was high. Also, the specific fuel type and chemical composition were not available as the measurements were conducted remotely."

References:

Fridell & Salo, 2014, Winnes et al, 2020 and Järvinen et al., 2023 were added.

In the manuscript, it is shown that faster vessel speed correlates with lower BC emissions and the authors also state that out of the ships having service speeds over 15 knots, all but one ship was equipped with EGCS, thus majority of the fastest ships would be the ones with EGCS installed. If the mean emission factor for ships applying / not applying EGCS is calculated from all observed plumes, would the authors expect the mean observed speed / engine load for these ships to also be different or affect the conclusion made regarding the effect of the EGCS? If so, it could be interesting to compare the effect of EGCS between plumes measured from ships with similar speed / modelled engine load. Furthermore, have the authors considered whether other parameters such as engine size or building year of the engine could affect the lower BC levels from the ships with EGCS installed?

**Reply**: As discussed in the previous reply, we have revised our manuscript on the effect of EGCS to black carbon emissions. As the speed and modelled main engine load of vessels without EGCS did not vary enough to define load-based emission factors, no conclusions can be drawn on them and between EGCS and non-EGCS ships.

p16. L267-273 It is stated that in the study, the BC particle ageing was assumed to have only a minor effect on the optically derived BC concentration due to short time periods between release and detection of the plume. However, the exhaust from ships without EGCS and the ones equipped with EGCS is released into the atmosphere in different conditions compared to the EGCS equipped ships, as the exhaust cools and is exposed to humid conditions inside the EGCS which in theory could allow coating of the particles already in the EGCS. Discussion could be extended to whether any evidence could be drawn from this study on if the EGCS affected the optically derived BC concentration or the optical properties of the detected BC.

**Reply**: It is true that in theory particle coating is possible also inside EGCS. Coating may systematically lead to slightly too high BC concentration values and therefore have an effect on the accuracy of the results and increase the uncertainty. However, we can assume that coating of particles is not dependent on ship speed or engine load, and therefore the precision is not affected and conclusions are not biased. Also, the emission factors presented in this study are in line with those measured by the others (see Discussion) which indicate that the coating effect is small or not significant.

The text is revised: "Even though BC is an inert compound and its chemical composition is not expected to change, it can still get coated with other materials inside the EGCS or during ageing that affect the optical properties of BC particles." and in addition: "It is also justified to assume that coating of particles inside the EGCS is not dependent on ship speed or engine load, and therefore conclusions are not biased. Still, in the ECGSs the exhaust cools down and is exposed to humid conditions, which could increase the coating on BC particles in comparison to exhaust that is not treated with an ECGS. Increase in coating could lead to increase in optically measured BC concentration for ECGS plumes compared to plumes from ships without an ECGS. The enhancement in measured absorption (i.e., BC concentration) due to coating depends on the wavelength, the coating material, the size of the BC core, and the thickness of coating (Lack and Cappa, 2010). For example, for ambient aerosol at an urban measurement site in Barcelona, depending on the amount of coating material, the absorption was increased by 1.1 - 1.6 times at 880 nm (Yus-Díez et al., 2022). With the current data set in this study, it is not possible to estimate whether the absorption was enhanced significantly with the plume aging or between plumes from ships equipped with or without an ECGS."

Technical comments:

p1. L10 Would propose using 'constant emission regime' if authors mean certain speed range, not to confuse with emission control areas (ECAs)

**Reply**: The sentence in the abstract edited to: "However, as calculated per hour the black carbon emission increased as a function of ship speed reaching peak values at around 15-20 knots and decreasing thereafter."

p2. L25 In addition to the division to climate warming and air quality effects of the air emissions, the authors could also mention the environmental effects or air emissions

**Reply**: The sentence has been revised to:

"Air emissions from ships can be categorised roughly by their impact on global warming, air quality and the environment."

Also, the environmental effects or air emissions are mentioned further in the paragraph:

"For example, fuel sulfur content limits have lead into increased harmful discharges into the sea (Jalkanen et al., 2024) and uptake of liquefied natural gas as shipping fuel into increased methane emissions (Lindstad & Rialland, 2020). California has successfully implemented voluntary speed limits in the effort of mitigating air pollution from sea-going vessels (Linder, 2018), as many pollutants such as nitrogen oxides (NO_x) and particulate matter (PM) are

associated with detrimental health and environmental effects (Chen and Hoek, 2020; Nunes et al., 2020; Orellano et al., 2020; Viana et al., 2020, Wang et al., 2020, 2019; Zhang et al., 2021)"

p2. L35 While in the study Lepistö et al. association was made between BC and lung-deposited surface area of particles, thus the potential of ship originated BC to introduce coemitted surface substances to the human lungs, I believe no health outcomes were shown directly. Probably other studies (e.g. Global health burden of ambient PM2.5 and the contribution of anthropogenic black carbon and organic aerosols - ScienceDirect) could be referred in regard to the ship originated BC health endpoints

**Reply**: This part was revised as:

" Black carbon has also been shown to contribute significantly to the health burden caused by fine particulate matter (Chowdhury et al., 2022) and recently an association was made also between lung deposited surface area and BC from ships (Lepistö et el., 2022)."

p10 L171 CO2 equivalent global warming potential. In later works, 100-year value of 900 has been presented for BC, the authors could justify the reasoning for using the 680 from the work of Bond and Sun (2005).

**Reply**: We have revised this and used more recent global mean black carbon GWP estimates of 1600 and 460 as per Fuglesvedt et al, 2010 and Gasser et al., 2017, which were added to the list of references. Lines 175-177 were revised to:

"Vessel total greenhouse gas emissions were calculated using 20 and 100-year global mean warming potential (GWP20 and GWP100) of BC, estimated as 1600 and 460 (Fuglesvedt et al., 2010, Gasser et al, 2017) in combination with vessel CO2 emissions to determine the effect of speed change in the total GHG as CO2 equivalent (CO2e)."

and also:

"CO2e emissions from black carbon represent on average 15.5% of total greenhouse gases using GWP20 and 5.2% using GWP100. As the total greenhouse gas emissions are dominated by CO2, they are reduced non-linearly with a reduction of speed (Figure 7)."

and also Figure 7 was updated with corresponding values. This did not however change the results or conclusions.

p 16 L261 In some previous literature, 'freshly emitted exhaust' or 'fresh exhaust aerosol' is used to depict exactly the exhaust aerosol few seconds or minutes after being released to surrounding atmosphere (ass measured from the plumes) whereas 'hot' or 'primary' aerosol is used to depict the exhaust aerosol as measured from the stack. Some clarification of wording is needed here

**Reply**: The text is clarified according to the reviewer's suggestion. The chapter reads now: "This so-called lensing effect can lead to an overestimation of BC concentration, when BC is derived by optical methods and a constant mass absorption cross section (for primary aerosol) is used."

References:
Fridell, E., & Salo, K. (2014). Measurements of abatement of particles and exhaust gases in a marine gas scrubber. Proceedings of the Institution of Mechanical Engineers, Part M: Journal of Engineering for the Maritime Environment, 230(1), 154–162. https://doi.org/10.1177/1475090214543716

Järvinen, A., Lehtoranta, K., Aakko-Saksa, P., Karppanen, M., Murtonen, T., Martikainen, J., Kuusisto, J., Nyyssönen, S., Koponen, P., Piimäkorpi, P., Friman, E., Orasuo, V., Rintanen, J., Jokiluoma, J., Kuittinen, N., & Rönkkö, T. (2023). Performance of a Wet Electrostatic Precipitator in Marine Applications. Journal of Marine Science and Engineering, 11(2). https://doi.org/10.3390/jmse11020393

Winnes, H., Fridell, E., & Moldanová, J. (2020). Effects of marine exhaust gas scrubbers on gas and particle emissions. Journal of Marine Science and Engineering, 8(4). https://doi.org/10.3390/JMSE8040299

**Reviewer 2:**

Heikkilä et al. discuss $CO_2$ and black carbon emissions from the ship plumes based on the measurements at a remote island in Finland. By presenting the dependence of $CO_2$ and black carbon emissions on the speed of the ship, the authors study the impact of speed limitation on climate and local air quality. This work focuses on ships with an exhaust gas cleaning system (EGCS), which seems to be common in this region. Hence, I find this work relevant to both the scientific community and to local society which may be interested in limiting ship speed. The paper is well-written and well-executed and its measurements and methods are well-documented. Since the study is based on ambient measurements, the distribution of measured plumes between different ship types is not uniform. That makes the statistical analysis more difficult to interpret. Therefore, I wish to see more discussion on the implications of that bias on the outcome of this paper. Hence, I have attached below my comments.

General comment:

My main comment concerns the statistical analysis and the fact that measurements are not uniformly divided between different ship types and that there exists a relationship between the ship type, ship speed, and having EGCS installed. I have listed below instances in which I think these possible biases should be discussed or taken into account.

I am slightly confused by the usage of the mean for emission factors in the result section. Based on the values of means and standard deviation as well as the measurement points presented in Figure 5, the data points are not normally distributed. I suggest using a median or a geometric mean. As far as I know, geometric mean has been used previously with emission factors, for example, Buffaloe et. al. 2013 and Schlaerth et. al. 2021

I would remove the tug from any statistical analysis in subsections 3.1 and 3.2 as it stands out from all other measurements and it may be not possible to determine whether its emission factor differs partly because the tug is 25-45 years older than the majority of analyzed ships. While using mean values for emission factors in statistical analysis, this one point of

measurement can drive mean values. However, I would still present the tug and its emission factor in Tables 1 and 2 as it is important to keep in mind that you have observed this case.

**Reply:** We appreciate for pointing this out. The distributions are truly not normal and therefore arithmetic means are not appropriate for comparison. We have updated Table 2 to include besides the arithmetic mean, but also the geometric mean and the median. Also, the statistical test between groups was changed from Welch two sample T-test to the Mann-Whitney U-test, which does not assume normal distributions. With the revision, median and geometric mean values are compared between groups and previous studies, and the Tug 1 plume does not therefore need to be excluded. The plume of Tug 1 isn't even a clear outlier: a plume generated by General cargo ship 7 had a BC emission factor of 3.49 g/kg fuel.

We have updated section 3.1 as follows:

"The fuel-based emission factors g BC (kg fuel)^-1) for vessels equipped with an Exhaust Gas Cleaning System (EGCS) were significantly lower (arithmetic mean: 0.22, geometric mean: 0.17, median: 0.15, standard deviation: 0.21) than for vessels without EGCS (AM: 0.99, GM: 0.82, MED: 0.83, SD: 0.68). The statistical significance in the difference of calculated emission factors between EGCS-equipped and vessels without EGCS was confirmed with the Wilcoxon Signed-rank test ($p < 0.01$). There is no statistically significant difference ($p = 0.06$) between BC emissions of ships in ballast condition (AM: 0.50, GM: 0.32, MED:0.29, SD: 0.50) and ships in laden condition (AM: 0.44, GM: 0.24, MED: 0.18, SD: 0.63). The fuel-based emission factors for ships with service speeds > 15.0 knots were significantly lower ($p < 0.01$) (AM: 0.25, GM: 0.18, MED: 0.15, SD: 0.24) than ships with service speeds $\le$ 15.0 knots (AM: 1.06, GM: 0.88, MED: 0.86, SD: 0.71). Vessel type BC emission factors are presented in Table 2 and in Figure 5."

And the Discussion as follows:

"The average BC emission factors (arithmetic mean: 0.48, geometric mean: 0.28, median: 0.24 and standard deviation: 0.56 g BC (kg fuel)^-1) derived from the aethalometer measurements for all examined ship exhaust gas plumes and vessel type-specific means Table 2 are in line with the results from previous literature using various measuring methods: Schlaerth et al., 2021 measured 78 plumes using a custom-built light absorption photometer obtaining a geometric mean emission factor of…"

and:

"Buffaloe et al., 2013 measured 91 ship plumes from onboard a research vessel combining multiple techniques (photoacoustic spectrometer and particle soot absorption photometer for light absorption, laser-induced incandescence and mass spectrometry) obtaining a geometric mean for all ships of.."

and:

"Lack et al., 2008 measured 101 ship plumes from onboard a research vessel using a photoacoustic technique to calculate the light absorbing carbon obtaining mean emission factors…"

and Conclusions:

"The median mean black carbon emission factor EF_BC for 47 ships representing 10 different vessel types measured from a remote marine station was 0.24 g BC (kg fuel)^-1. For ships equipped with EGCS it was 0.15 and for ships without EGCS 0.83."

While discussing the difference between ships with and without EGCS, there should be a mention that certain ship types belong only or almost only to one category. If table 1 will be changed as I suggested below that should be easy to notice from that table. I would also love to see a short discussion on whether this imbalance could affect the outcome of statistical analysis. For example, all Roro have EGCS equipped and their plumes count for 137 out of 145 (if I count correctly) plumes analyzed as coming from a vehicle equipped with EGCS. Is the BC emission factor of vessels equipped with EGCS significantly different than the one for Roro? Please, discuss that.

**Reply**: Based on this and another reviewer's comment, we have edited the part in the Discussion concerning vessels equipped with EGCS as follows:

"Ships without EGCS had a median BC emission factor 5-fold larger than vessels with EGCS. Previous literature confirms that EGCS reduces particle mass and BC output (Lack et al, 2012, Fridell  and Salo, 2014, Lehtoranta et al., 2019). However, in Winnes et al (2020)  and Järvinen et al. (2023) BC output was higher when combusting HFO in combination with EGCS compared to combusting low-sulfur fuel oil without EGCS. The modelled engine load for the ships without EGCS was relatively low in this study, which could explain why their averaged BC output was high. Also, the specific fuel type and chemical composition were not available as the measurements were conducted remotely."

In the result section (3.1) a statistical difference in BC emission factor for vehicles with EGCS and without it as well as a statistical difference in BC emission factor for vehicles faster and slower than 15 knots are stated. Further we can read in the text: 'All except for one vessel passing the measuring site with a speed > 15 knots over the ground were equipped with EGCS. Most vessels without EGCS were also ships with lower service speeds and therefore would not need to slow down for the pilot exchange.' Hence, while grouping data between ships by having EGCS or by having a service speed higher or lower than 15 knots, almost identical division is made. However, the way how the data is presented and described implies that having EGCS and service speed are independent from each other and that is not true. I think this issue needs to be addressed while presenting these BC emission factors and a short discussion should be added on which of these two properties (speed or EGCS) is expected to have a bigger impact and why. Additionally, the box plots of BC emission factor log10 measured from passing ships grouped by having an Exhaust Gas Cleaning Systems (Figure 5 top left) and having a service speed larger or less than 15.0 knots (Figure 5, bottom right) are almost the same due to the issue described above.

**Reply**: In our dataset grouping ships by having EGCS equipped or not is done on purpose as previous literature shows that EGCS reduces particles and also black carbon in the exhaust gas. It would have been wrong in our opinion to calculate a load-based emission factor function including datapoints from both groups. We would have wanted to be able to calculate it for both groups separately and compare them, but our dataset did not support this. There were 4 ships and 8 plumes in our dataset of ships without EGCS and service speed at or above 15.0 knots so these groups are not identical. Also, the service speed limit is quite artificial instead of having equipment that is known to change the characteristic of the exhaust gas.

The correlation between BC emission factor and speed over ground was presented for all vehicles, vehicles without EGCS, and vehicles with EGCS. Is there a statistical difference between measurement for all vehicles and the vehicles with EGCS (145 out of 211 plumes). If yes, please report it. If not, please do not present both values.

**Reply**: The correlation analysis was revised to:

"Vessel speed over ground correlates negatively with the BC emission factor. Pearson's correlation between speed and BC emission factor on ships with EGCS was -0.60 (95% confidence interval -0.70 - -0.49, p > 0.01) and on ships without EGCS it was -0.32 (95% confidence interval -0.51- -0.09, p = 0.01)."

I would also add a brief discussion on whether the regression line for log10 BC emission factor as a function of weather adjusted main engine load serves well for cruise and ropax and whether there is an uncertainty in calculating load-based emission factor worthy of reporting based on that fit for these ship types.

**Reply**: The following was added to Section 3.3.:

" The EGCS dataset was dominated by plumes measured from roro-type vessels (137 observations) and the adjusted $r^2$ for roro-type vessels only was 0.39. Two plumes were observed from EGCS-equipped cruise vessels, each from a different vessel. For vessel Cruise 3 the model fits well (observed EF_BC: 0.21, predicted 0.25) but poorly for vessel Cruise 2 (observed EF_BC: 0.09, predicted: 0.29). Three plumes were observed from EGCS-equipped vessel Ropax 1 on which the model predicts well on higher engine load (observed EF_BC: 0.11 and 0.15, predicted: 0.13 and 0.14) but poorly with lower engine load: (observed: 1.07, predicted: 0.31)."

and the following to the Discussion:

"The load-based BC prediction model performed reasonably and some outliers could be explained by the ship momentarily slowing down for pilot boarding or disembarking and having more engines online than would be optimum for the speed."

Minor comments:

Figure 2: I suggest coloring the bottom subplot (ship speed vs time) by the measured concentration of $CO_2$ or BC. Currently, the plot is colored by the time which doubles that information on the figure. I think following the evolution of the plume in $CO_2$ or BC data will be a worthy addition to this example and it would clarify the meaning of 'plume start time'.

**Reply**: Figure modified as recommended also taking into account the recommendation just below. Caption was modified accordingly: "The lower panel shows the ship speed, observed CO$_2$ concentration. The shaded area denotes the time that was used for defining the pollution background levels and the darker shading denotes the time of the actual plume was observed."

Line 97: 'We applied a method introduced by Ausmeel et al. (2019) to calculate the background, which was defined as the median value of 6 minutes before the plume started and 6 minutes after the plume ended omitting the period of the plume.' – I suggest visualizing that by marking the time used for the background (for example by shading the area) on the bottom subplot of Figure 2.

**Reply**: See reply above. Added a reference to Fig. 2 in the end of the sentence: "(example in Fig. 2).."

Table 1: I think that table one is necessary for this paper, but I suggest moving it to an appendix or supplementary material. I suggest using in the main text a table giving an overview of that information, for example for cruises:

| N | Vessel | ME | PR | kW | BY | SS | PL | ES | DE | SG |
|---|--------|-----|-----|-----------|-----------|-----------|-----|-----|-----|-----|
| 3 | cruise | 4-5 | 2 | 32000-55216 | 1993-2020 | 20.0-22.5 | 5 | 2 | 3 | 0 |

The overview gives a better introduction to the statistical analysis chapter.

**Reply**: Table 1 changed according to suggestion and longer table moved to Appendix A.

Figure 7: Would it be worth to also present total greenhouse gas emissions per nautical mile versus speeds? To consider the long-term effect of greenhouse emissions it is perhaps better to compare it by distance.

**Reply**: Figure 7 plots changed to CO2e/NM.

Technical comments:

Line 25-26: 'In many cases, focusing only on one could have a negative impact on the other. ' - I would add references for articles showing such cases.

**Reply**: The following has been added to the introduction:

" For example, fuel sulfur content limits have lead into increased harmful discharges into the sea (Jalkanen et al., 2024) and uptake of liquefied natural gas as shipping fuel into increased methane emissions (Lindstad and Rialland, 2020)."

and the following were added to the list of references:

Lindstad, E.; Rialland, A. LNG and Cruise Ships, an Easy Way to Fulfil Regulations—Versus the Need for Reducing GHG Emissions. *Sustainability* **2020**, *12*, 2080. https://doi.org/10.3390/su12052080

Jalkanen Jukka-Pekka et al., Environmental impacts of exhaust gas cleaning systems in the Baltic Sea, North Sea and the Mediterranean sea area, Tech Report, EMERGE Project, http://doi.org/10.35614/isbn.9789523361898, 2024.

-

Line 102: 'However, in this application, the method was not optimal due to rather rapid changes in the background levels' – I find this sentence not as clear as it could be. In this paragraph different methods are discussed, also for background calculation, hence it is slightly ambiguous which 'this application'/'method' you mean.

**Reply**:This part was modified to be more exact: "In our study, where the plumes originated from a relatively short distance (about 2 km) and were short in duration (about 5 min), the method by Ausmeel et al. (2019) produced more well-defined plumes as it took variation in background concentrations better into account. The method by Kivekäs et al. (2014) suits

automatic plume detection, whereas the method by Ausmeel et al. (2019) requires more manual work."

Line 108: You are using acronyms HFO and MGO which are explained later in the text (in lines 116-117). Please, explain acronyms with their first usage.

**Reply**: Acronyms corrected

-

Line 115 and 123: 'an exhaust gas cleaning system (EGCS)' and 'an exhaust gas cleaning system (ES)' – please use only one acronym.

**Reply**: Acronym corrected to EGCS in Table 1.

-

Table 2: In Table 2 there are listed chemical tankers and product tankers while Table 1 contains only tankers category. Please be consistent with your categories.

**Reply**: Table corrected with Chemical and Product tankers identified.

-

Line 230-231: 'Total greenhouse gas emissions are dominated by CO2 and are reduced non-linearly with a reduction of speed (Fig. 7).' –I would add a small explanation to that statement (for example, 'Since the total greenhouse gas emissions…') so it is easier for the reader to understand it where it is coming from. It is the only sentence in this chapter (possibly in the result section) that discusses mainly $CO_2$, not BC. I would either add a sentence or two to make that transition easier for the reader or re-write this statement from BC perspective.

**Reply**: The paragraph was edited as follows:

"$CO\_2e$ emissions from black carbon represent on average 15.5% of total greenhouse gases using $GWP\_20$ and 5.2% using $GWP\_100$. As the total greenhouse gas emissions are dominated by $CO\_2$, they are reduced non-linearly with a reduction of speed (Figure 7).

Sometimes the references for figures in the text are as 'Figure' sometimes as 'Fig.'. Please, unify it.

**Reply**: unified to Figure throughout the manuscript.

The colors used throughout the paper are sometimes hard to differentiate (for example, Ropax and Roro2 in Figure 7) and I am not sure if they are colorblind-friendly. I would consider changing the color scale. Additionally, in Figure 1 (bottom left plot) the markers for the bulk carrier and the tug are covered with the color markers. Hence, I would either change the marker size or type for these plots.

**Reply**: Figure 7 replotted using colorblind-friendly colors as per Okabe & Ito (2008).

Please, check your reference. For example, the link to Buffaloe et. al. does not lead to the original print.

**Reply**: References checked throughout and the correct DOI included in Buffaloe et al., 2014

References:

- Buffaloe, G., Lack, D. A., Williams, E. J., Coffman, D. J., Hayden, K., Lerner, B. M., Li, S.-M., Nuaaman, I., Massoli, P., Onasch, T. B., 345 Quinn, P. K., and Cappa, C. D.: Black carbon emissions from in-use ships: a California regional assessment, Atmospheric Chemistry and Physics, 14, 1881–1896, https://acp.copernicus.org/articles/14/1881/2014/, 2013.
- Schlaerth, H., Ko, J., Sugrue, R., Preble, C., and Ban-Weiss, G.: Determining black carbon emissions and activity from in-use harbor craft in Southern California, Atmospheric Environment, 256, 118 382, https://doi.org/10.1016/J.ATMOSENV.2021.118382, 2021

---

## Author Response (AR2)

**Reply to reviewer**:

The manuscript has been improved during the revision process and the authors have made sufficient changes to respond to reviewer comments. I encourage accepting the manuscript with following minor revisions.

**Reply**: The authors would like to thank the reviewer for an excellent review and we believe the manuscript has truly benefited from the suggested changes.

Section 3.1.:
The authors have improved the section discussing black carbon emission factors. As commented before and pointed out by the second reviewer, the ship service speed and EGCS (and also ballast / laden condition) are not independent groups. I would suggest the authors to show additional figures where the data points shown in Figure 5 are shown in such a manner that the service speed is in the x-axis and BC emission on the y-axis and circles or color coding is used to indicate the data points of ship with or without EGCS. (Similar figure could be shown for ballast / laden condition). This might perhaps accompany the table A1 or be two additional panels in Figure 5.
The additional figure would provide the reader a better understanding of the underlying data behind the statistical analysis and dependency between service speed and EGCS in the observations.

**Reply**: We have added the Figure B1 in the Appendix B as suggested by the reviewer. Also, we have added the following sentence to the Section 3.1.:

"$EF_{BC}$ as a function of ship service speed between different loading conditions and EGCS is presented in Figure B1 of Appendix B."

Conclusions L339-340:
For the conclusion regarding BC emission from ships with EGCS, I would encourage to modify the sentence to include the remark that majority of EGCS equipped ships also had faster service speed.

**Reply**: We have added the following sentence to the Conclusions as suggested by the reviewer:

"The majority of vessels equipped with EGCS also had faster service speeds."